# Low-Input Crops as Lignocellulosic Feedstock for Second-Generation Biorefineries and the Potential of Chemometrics in Biomass Quality Control

**Abla Alzagameem [1,2], Michel Bergs [1], Xuan Tung Do [1]**, **Stephanie Elisabeth Klein [1],**
**Jessica Rumpf [1], Michael Larkins [3], Yulia Monakhova [4,5,6], Ralf Pude [7] and Margit Schulze [1,\*]**

[1]  Department of Natural Sciences, Bonn-Rhein-Sieg University of Applied Sciences, von-Liebig-Strasse 20, 53359 Rheinbach, Germany; abla.alzagameem@h-brs.de (A.A.); michel.bergs@h-brs.de (M.B.); xuan-tung.do@h-brs.de (X.T.D.); stephanie.klein@h-brs.de (S.E.K.); jessica.rumpf@h-brs.de (J.R.)

[2]  Faculty of Environment and Natural Sciences, Brandenburg University of Technology BTU Cottbus-Senftenberg, Platz der Deutschen Einheit 1, D-03046 Cottbus, Germany

[3]  Department of Forest Biomaterials, North Carolina State University, 2820 Faucette Drive Biltmore Hall, Raleigh, NC 27695, USA; mclarki2@ncsu.edu

[4]  Spectral Service AG, Emil-Hoffmann-Strasse 33, D-50996 Köln, Germany; yul-monakhova@mail.ru

[5]  Institute of Chemistry, Saratov State University, Astrakhanskaya Street 83, 410012 Saratov, Russia

[6]  Institute of Chemistry, Saint Petersburg State University, 13B Universitetskaya Emb., 199034 St Petersburg, Russia

[7]  Field Lab Campus Klein-Altendorf, Faculty of Agriculture, University of Bonn, Campus Klein-Altendorf 1, D-53359 Rheinbach, Germany; r.pude@uni-bonn.de

\*  Correspondence: margit.schulze@h-brs.de; Tel.: +49-2241-856566



**Featured Application: 1. The utilization of so-called low-input crops (i.e., *Miscanthus* grasses and fast-growing trees) as lignocellulosic feedstock for second generation biorefineries. 2. Lignin and lignin-derived materials as agrochemical products. 3. Chemometric methods to be used for fast and efficient lignocellulose feedstock (LCF) quality control.**

**Abstract:** Lignocellulose feedstock (LCF) provides a sustainable source of components to produce bioenergy, biofuel, and novel biomaterials. Besides hard and soft wood, so-called low-input plants such as *Miscanthus* are interesting crops to be investigated as potential feedstock for the second generation biorefinery. The status quo regarding the availability and composition of different plants, including grasses and fast-growing trees (i.e., *Miscanthus*, *Paulownia*), is reviewed here. The second focus of this review is the potential of multivariate data processing to be used for biomass analysis and quality control. Experimental data obtained by spectroscopic methods, such as nuclear magnetic resonance (NMR) and Fourier-transform infrared spectroscopy (FTIR), can be processed using computational techniques to characterize the 3D structure and energetic properties of the feedstock building blocks, including complex linkages. Here, we provide a brief summary of recently reported experimental data for structural analysis of LCF biomasses, and give our perspectives on the role of chemometrics in understanding and elucidating on LCF composition and lignin 3D structure.

**Keywords:** chemometrics; lignin; lignocellulosic feedstock; low-input crops; multivariate data analysis; *Miscanthus*; *Paulownia*; *Silphium*

## 1. Introduction

Global economic and ecological challenges of the twentieth century, such as limited fossil resources, climate change due to greenhouse gas emissions, and the global energy demand, are driving forces for

innovations in chemical industry. Facing these challenges, the European Bioeconomy Strategy was first reported by the European Commission in 2012, and updated in 2018 [1,2]. In total, the annual turnover of the European bioeconomy was estimated to be €2.3 trillion, involving about 18.5 million people, including biorefineries of the first- and second-generation. Compared to the first-generation concepts, the second-generation biorefineries do mainly focus on non-food crops and wastes from agroforestry. According to the authors of a European study, about 476 million tons of lignocellulose feedstock (LCF) is required to fulfil the demand for bio-based products by 2030. Today, more than 70 lignocellulosic biorefineries (mainly pilot plants) have been established for LCF exploitation [3]. According to studies performed by the Food and Agriculture Organization (FAO), about 70 million ha of additional cultivated land will be required by 2050 for feed and food production [4]. The driving force to study the potential of renewable resources, in particular lignocellulose feedstock, is the development of novel bio-based materials, such as polyol components, for polyurethane synthesis. Besides polyols produced from vegetable oils, lignin is studied as a substitute for fossil-based diols and polyols (Figure 1) [5,6].

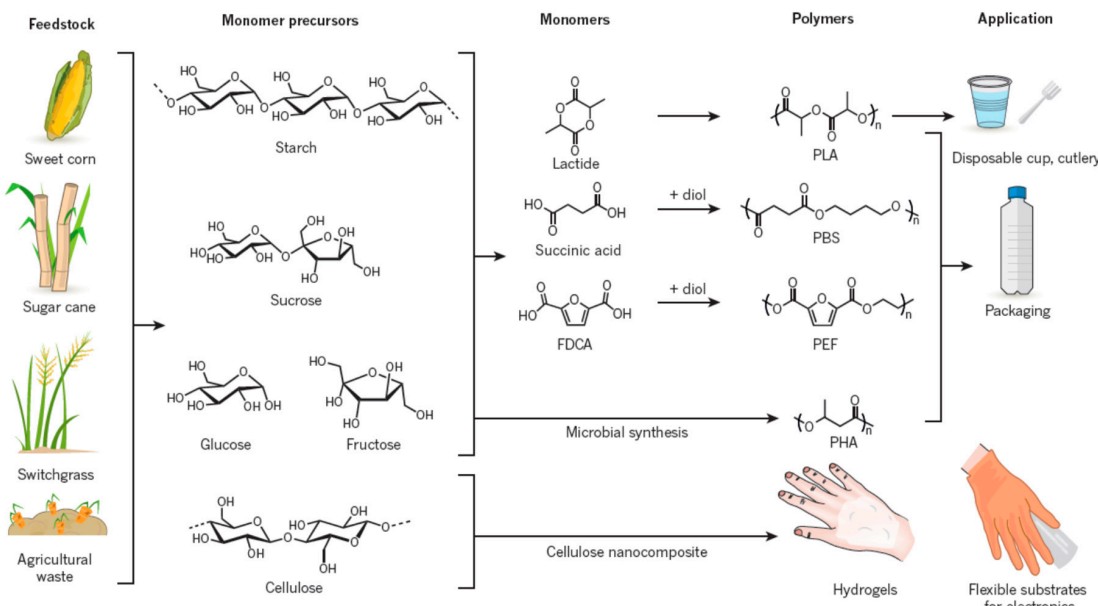

**Figure 1.** Sustainable polymers from polysaccharides and agricultural waste. Reproduced with permission from [5], Springer Nature, 2019.

## 2. Lignocellulose Feedstock Biorefineries

### 2.1. First- and Second-Generation LCF Biorefineries

For the first-generation biorefineries, sugar and starch crops were used (i.e., sweet corn, sugar cane) [7–12]; there was then a change in feedstocks in the so-called second-generation biorefinery—potential biomasses for these new refineries include grasses cultivated in arid conditions, agroforestry residues, and any kind of crop waste (Figure 2) [13–16].

Statistics show that 170 million metric tons of lignocellulose is produced annually, while no more than 5% of these LCF components are exploited, mainly due to a significant recalcitrance caused by the lignin [17]. Biorefining is an important option to carry out innovative valorization of lignocellulosic materials, which has triggered intense research on how to convert lignins into target chemicals and fuels. LCF sources for biorefinery use include soft and hard wood, lignocellulose-rich grasses, and agroforestry waste. The market for bio-based products is expected to increase to €50 million by 2030 (average annual growth rate of 4%) [13].

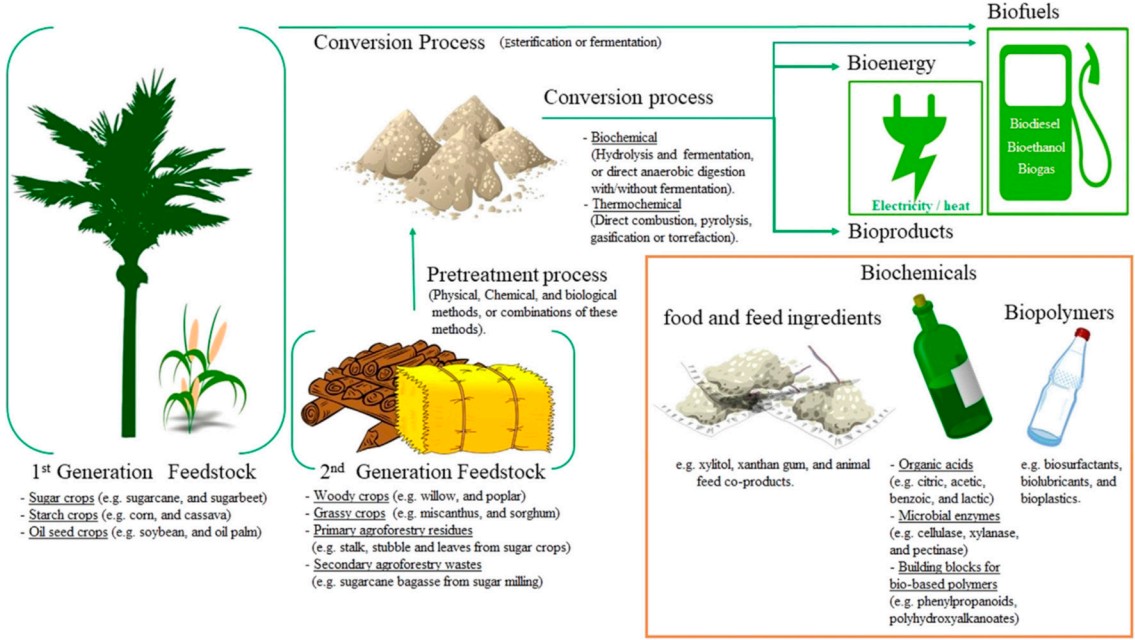

**Figure 2.** Schematic diagram shows the differences between lignocellulosic feedstocks from the first and second generations: sources, valorization processes, and end products. Reproduced with permission from [13], Elsevier, 2019.

According to a recently reported market study, until 2023 an annual growth rate of 2% is predicted for the global lignin market, resulting in an increase of the total market size from €800,500,000 in 2017 to €904,500,000 in 2023 [18,19]. Among the most interesting products generated from lignocellulosic biomasses are biofuel and bioethanol. Here, we focus the isolation and application of lignins obtained from LCF biomasses. Lignin is mainly studied as a polyol-substitute for polyurethane and resin production, but also as an electrode material for sustainable electrochemical energy storage [20].

Lignocellulosic biomasses are rather resistant to enzymatic and chemical hydrolysis and therefore require harsh reaction conditions (i.e., strong acids or bases). LCF pretreatment and pulping results in the separation of cellulose/hemicellulose and lignin. Depending on the pulping process, the macromolecular lignin is partially degraded. In their review articles, Rinaldi et al. and Schutyser et al. discussed lignin depolymerization strategies (catalyzed reductive and oxidative cleavage, respectively) and correlated mechanisms in order to produce lignin oligomeric fragments, such as phenol derivatives, to be used for further polymerization [21,22].

In general, the detailed 3D lignin structure (monolignol ratio and linkages) depends on a number of different parameters: the biomass source and crop genotype/phenotype, due to different biosynthesis pathways (i.e., soft and hard wood, grasses), and the pulping process (e.g., kraft, steam explosion, organosolv). Figure 3 shows the most common lignin linkages formed during biosynthesis, some of which having been elucidated within the last five years [23,24]. Table 1 shows average values for monolignol linkages found for hard/soft wood and grasses [25–27]. These structural differences are rather difficult to specify by conventional analytical methods using data univariate processing, due to signal overlapping in spectral data.

Hirayama et al. studied the ratio of biphenyl fragments (5–5′ linkages) of different biomasses (six softwoods and 15 hardwoods) [28].

A focus of lignin-derived materials includes novel bio-based polymers, such as polyurethanes [29–34], as coatings and/or foams for construction applications. In addition, the bioactivity of lignins is widely studied, including antioxidant, antiviral, and antimicrobial activity [35–38]. In order to obtain valuable oligomer fragments, the macromolecular lignin structure is depolymerized using various strategies, including oxidative and reductive depolymerization via homo-

and heterogeneous catalysis, ozonolysis, and photolysis [21,22,39,40]. Very recently, Renders and colleagues reported the concept of a so-called "lignin-first biorefinery", which is based on a reductive catalytic fractionation (RCF) of lignocellulose biomass. The RCF procedure combines a lignin catalytic depolymerization with fractionation of the degraded low molecular weight lignin oligomers, or even monomers (i.e., alkylated catechols) [41].

**Figure 3.** Examples of monolignol linkages. **Top**: Ether bonds (β-O-4′, α-O-4′, 4-O-5′); **Middle**: C–C bonds (β-β′, β-1′, 5-5′); **Bottom**: Complex linkages (β-5′/α-O-4′, 5-5′/β-O-4′/α-O-4′, β-1′/β-O-4′). Reprinted from [27] under open access license.

**Table 1.** Abundance of linkages in lignins of soft and hard wood and *Miscanthus* grasses, including KOH-extractable and non-KOH-extractable, in percentages. Reprinted from [27] under open access license.

| Linkage | Hard Wood H/G/S traces/25-50/50–75 | Soft Wood H/G/S 0.5–3.4/90–95/0–1 | *Miscanthus* H/G/S 24/49/27 |
|---|---|---|---|
| β-O-4′ | 50–65 | 43–50 | 93 |
| A-O-4′ | 4–8 | 6–8 | ns* |
| β-β′ | 3–7 | 2–4 | 4 |
| β-5′ | 4–6 | 9–12 | 3 |
| β-1′ | 5–7 | 3–7 | traces |
| 4-O-5′ | 6–7 | 4 | ns* |
| 5-5′ | 4–10 | 10–25 | ns* |

ns*: not specified.

By 2023, the lignin market volume is expected to increase up to 18 million tons and US$6.0 billion [18,19]. In particular, the kraft lignin market volume will increase up to 125 kilo tons by 2021 and more than US$5 billion. For example, in North America the lignin market is dominated by lignosulfonates used as concrete and cement flow improver. Europe is the second largest market for lignin. Unlike North America, the focus is directed to lignin-based materials (end-use industry). The lignin market is segmented on the basis of product type, application type, and geographical analysis. By product type, this market is segmented on the basis of lignosulfonates, Kraft lignin, Organosolv lignin, and high purity lignin. Today, lignocellulose-rich biomasses, including agrochemical waste, are processed all over the world in commercial mills, demonstration plants, and pilot scale facilities, to produce pulp, paper, lignin, and various LCF-derived chemicals (Table 2) [42–61].

**Table 2.** Pilot plants and industrial production sites for lignocellulose feedstock (LCF) exploitation and valorization [42–61].

| Company/Institution | Location | Production Scale | Feedstock and Products | Reference |
|---|---|---|---|---|
| Borregaard LignoTech | Sarpsborg, Norway | Industrial scale | World leader in lignin-based products (lignins and lignosulfonates and lignin-derived chemicals). In Fernandina Beach, FL, USA: Southern yellow pine-based lignin utilizing a coproduct of RYAM's sulfite pulping process | [18,19,42–45] |
| Tembec/Rayonier Advanced Materials | Jacksonville, FL, USA | Industrial scale | Paper, pulp and lignin production | [18,19,46,47] |
| Domtar Corporation | Montreal, QC, Canada | Industrial scale | LignoBoost plant in Plymouth. Pine-based BioChoice® | [18,19,48–50] |
| Asian Lignin Manufacturing Pvt. Ltd. | Chandigarh, India | Industrial scale | Paper, pulp and lignin production | [18,19] |
| Northway Lignin Chemical | Sturgeon Falls, ON, Canada | Industrial scale | Paper, pulp and lignin production | [18,19,48] |
| GreenValue SA | Orbe, Switzerland | Industrial scale | Sulfur-free lignin. Feedstock: wheat straw, flax, sugar cane. Aqueous alkaline extraction. | [18,19,45,51] |
| Domsjö Fabriker AB (world's 2nd largest producer of powder lignin). Domsjö is part of the Aditya Birla Group. | Örnsköldsvik, Sweden/Aditya Birla Headquarter Mumbai, India | Industrial scale | Powder lignin. Domsjö is the world's 2nd largest producer of Lignin powder with its origin from sustainable Swedish forestry. | [18,19,42] |
| Changzhou Shanfeng Chemical Industry Co. Ltd. | Changzhou, Jiangsu, China | Industrial scale | Lignin polyether polyols | [18,19] |
| The Dallas Group of America | Whitehouse, NJ, USA | Industrial scale | Lignosulfonates | [18,19,46,47] |
| Nippon Paper Ind. Co. Ltd. | Tokyo, Japan | Industrial scale | Lignosulfonates | [18,19,45] |
| Liquid Lignin Company, LLC | Clemson, SC, USA | Industrial scale | Liquid Lignin Company develops and commercializes new lignin-based technologies. | [18,19,46,52,53] |
| Metsä Group | Espoo, Finland | Industrial scale | Forests and wood-based bioproducts | [18,19,45,54] |
| Fibria | Sao Paulo, Brasilia | Industrial scale | Forests and wood-based bioproducts. World's leader in Eucalyptus-derived pulp. | [18,19,45] |
| Lenzing AG | Lenzing, Austria | Industrial scale | Forests and wood-based bioproducts. European leader in pulp production. | [18,19,45] |
| Stora Enso | Helsinki, Finland | Industrial scale | LignoBoost plant at Sunila mill. Lineo™® (wood-based). Kraft pulping process of Nordic softwood, pine and spruce. The refined kraft lignin is available as a stable, free-flowing brown powder or a moist powder block. 50,000 tons of dry lignin per year. | [18,19,45,50,54] |
| Weyerhaeuser Company (in collaboration with Lignol Energy Corp./Fibria Cellulose SA) | Seattle, WA, USA | Industrial scale | Second generation biofuels and chemicals | [18,19,46–49] |
| GreenField | Boucherville, Quebec, Canada | Industrial scale | Biobased alcohols | [18,19,45,48] |
| Enchi Corp. | Lebanon, NH, USA | Industrial scale | Bioenergy and biofuels | [18,19,46,49] |
| Microbiogen | Lane Cove West, Australia | Industrial scale | Bioethanol and bioethanol producing yeast. | [18,19,55] |
| DuPont/VERBIO North America Corporation (VNA), Grand Rapids, Michigan, U.S. | Nevada, IA, USA | Industrial scale | Second generation biofuels and chemicals. World's largest cellulosic ethanol and biofuel facility (30 million gallons per year). Corn stover feedstock. | [18,19,46,56] |
| POET-DSM | Sioux Falls, SD, USA | Industrial scale | Second generation biofuels and chemicals | [18,19,56,57] |
| IOGEN Corp. | Ottawa, ON, Canada | Demonstration | Second generation biofuels and chemicals. Cellulosic Ethanol. Crop Residue Feedstock. | [18,19,48] |
| Fraunhofer Center for Chemical-Biotechnological Processes (CBP) | Leuna, Germany | Pilot plant | Wood-based organosolv lignin: debarked beech wood (Fagus sylvatica) chips by ethanol–water-pulping in a batch process (70 kg dry biomass). | [18,19,58,59] |
| Bioprocess | Delft, The Netherlands | Pilot plant | Biomass hydrolysis and fermentation | [18,19,60] |
| bioCRACK | Schwechat, Austria | Pilot plant | Second generation biofuels | [18,19,61] |

### 2.2. Reported Techno-Economic Analysis Studies

Currently, there are a number of techno-economic analysis studies reported including information about the economic value and environmental impact of single LCF products, such as bioethanol. For example, in 2019 Da Silva et al. published an assessment of different LCF pretreatment processes for bioethanol production. Taking into account five different pretreatment procedures of lignocellulosic biomass, the authors found that diluted acid is the best choice for bioethanol production, with an economic value of $39.2 million per year and an environmental impact of 83.9 kt $CO_2$ per year [62]. Patel et al. tried to quantify the production cost of biodiesel from agricultural waste, a comparative assessment recently reported [63]. Also in 2019, Albashabsheh et al. published their study on "mobile pelleting", a procedure applied to improve and optimize lignocellulosic biomass-to-biofuel supply chains. In particular, the authors investigated mobile pelleting machines (MPM) to minimize logistical costs and to find out at which point mobile densification becomes economically attractive. Therefore, they included about 20 different input parameters, like the type and price of biomass, densification and transport costs, storage capacity, and number of MPMs available [64]. A similar approach was reported by Srivastava et al. in 2019, to analyze costs for biofuel production [65].

In her PhD thesis, Karkee investigated the optimization and cost analysis of LCF supply chains. Considering corn stover as a by-product of grain production, the gate price of the biomass feedstock varies from $75 $Mg^{-1}$ to $97 $Mg^{-1}$ (depending on different factors, such as farm size, transport distance, and stover yield) [66]. Furthermore, the costs for harvesting and transport have been determined for different feedstocks (i.e., switchgrass). Quantification models were used which considered the number of machines, farm size, and biomass yields. Zhao et al. reported a Chinese market techno-economic analysis for the production of bioethanol. In particular, pretreatment using dilute acids and an enzymatic hydrolysis were studied for corn stover biomass. Using two different models, the authors calculated the plant-gate price for bioethanol and reported it to be $4.68–$6.05/gal following a biochemical conversion pathway. Thus, at this price point, ethanol from lignocellulose biomasses is still unable to compete with ethanol from fossil resources [67].

In their techno-economic analysis study reported in 2011, Gnansounou et al. comprehensively reviewed data for ethanol production from lignocellulosic feedstocks. They could identify and quantify some key parameters influencing the production costs, like type and composition of feedstock and its farm-gate price, conversion efficiency, the ethanol plant size, and the extent of investment costs, using three different types of cost management system, whereby the most significant contribution to the overall lignocellulosic bioethanol production costs is the biomass cost [68].

### 2.3. Low-Input Crops: Sources and Availability

#### 2.3.1. $C_4$ Grasses: Miscanthus

According to the European Common Agricultural Policy regulations, there are three so-called "greening measures", including maintenance of permanent pastures, crop diversification, and ecological focus areas (EFA) [69]. Thus, 5% of the land has to be specified as EFA by European farmers. Very recently, *Miscanthus* (an analogue to other perennial crops, such as short rotation coppice) was listed as an eligible EFA crop. *Miscanthus* genotypes combine different advantages, such as biodiversity and a significant greenhouse gas emission reduction [70–74]. In 2019, John Clifton-Brown et al. reported a detailed study of the breeding progress of various lignocellulose-rich biomasses, including switchgrass, *Miscanthus*, willow, and poplar crops [75,76].

Bergs et al. studied both the crop composition and detailed chemical structure of the corresponding *Miscanthus*-derived lignins. In detail, harvest yields of six different *Miscanthus* genotypes have been studied and compared for the years 2015 and 2016 [26,27]. Here, *M. nagara* showed the highest yields compared to various *M. x giganteus* samples, with *M. robustus* and *M. sinensis* having lowest values of all different genotypes.

*Miscanthus* crops belong to the group of perennial $C_4$ plants. Unlike $C_3$ plants, which produce D-3-phosphoglycerate, $C_4$ plants generate oxaloacetate, which is correlated with a significant effect on carbon sequestration [77–79]. Due to a rather low level of required water and fertilizer they are called low-input crops [80–82]. Figure 4 shows fields with different *Miscanthus* genotypes, cultivated at the Campus Klein-Altendorf in Rheinbach, Germany. *Miscanthus* crops are rather tall (up to four meters), yielding up to 25 t/ha.

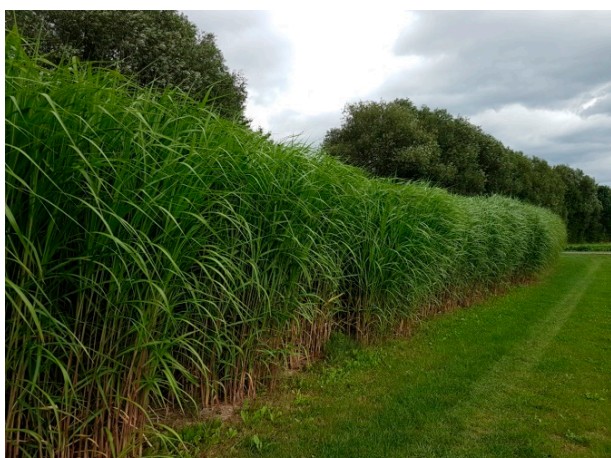 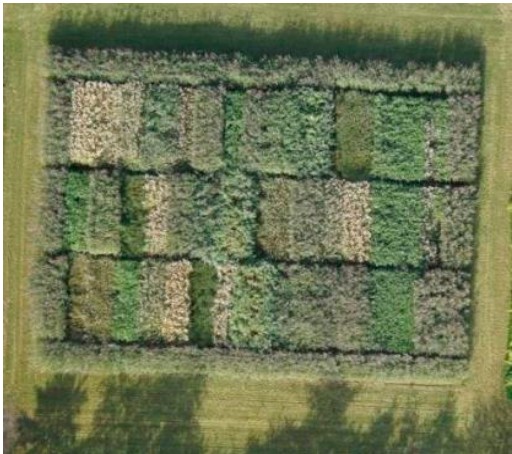

**Figure 4.** *Miscanthus* fields at the Campus Klein-Altendorf, Rheinbach, Germany. **Left**: *Miscanthus x giganteus* in spring time. **Right**: Color differences resulting from crops planted in different seasons; light colored parcels of dried crops versus darker fields of freshly planted crops. Copyright 2019, Katharina Walbrück.

The advantages of perennial plants in general are rather low production costs, due to less tillage [83–86]. Kraska et al. recently reported the cascade utilization of *Miscanthus*, including exploitation of the stalks and fibers, as well as the leaves [87,88]. Other research groups reported the utilization of *Miscanthus* crops for the production of bioethanol [89], hydrogen [90], and other chemicals, including polymers and composites [91–95]. Although there is a huge number of published studies, very few systematic studies are available about *Miscanthus*-derived lignins [96–103]. Van der Weijde determined the cell wall composition of eight different M. *sinensis* samples [104]. Various authors reported the enzymatic depolymerization of *Miscanthus*-derived lignins, such as Baker, Ion, and Sonnenberg [105–107]. However, all of these studies exclusively focused on crop composition analysis (lignin ratio and distribution), but no details were reported regarding the detailed lignin structure.

2.3.2. Fast Growing Trees: Paulownia, Eucalyptus, and Pinus

Due to recent efforts in biorefinery development, fast growing trees attract more and more attention for study as an industrial crop. Besides bamboo, poplar, Eastern cottonwood, giant sequoia, and acacia (not discussed here), Eucalyptus, pine, and *Paulownia* belong to the fast growing lignocellulose-rich crops that are currently under investigation to be used as potential feedstock for second-generation biorefineries. Compared to conventional trees, the growing cycles (silviculture rotations) of fast-growing trees are below 15 years, thereby offering environmental and/or genetic manipulation [108].

One prominent example is the fast-growing *Paulownia* tree, originally cultivated in Asia, mainly in China and other tropical and sub-tropical regions, and characterized by a low demand for water. *Paulownia* trees grow quickly, reaching 10 to 20 m in height and 30–40 cm in diameter in less than ten years. Ye and colleagues reported a study on *Paulownia tomentosa*, a genotype that reaches 30–40 cm in diameter within five years (Figure 5) [109].

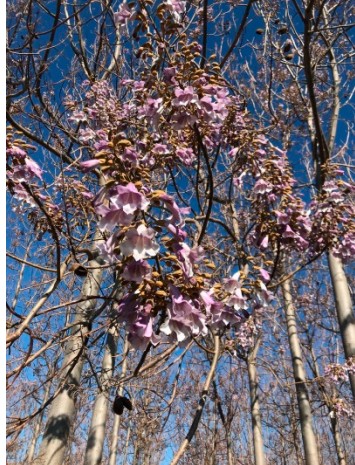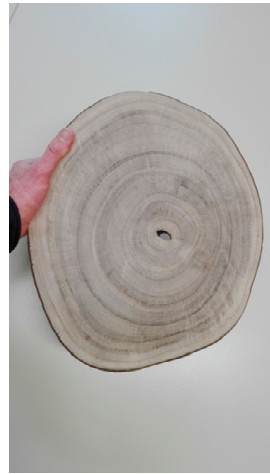

**Figure 5.** *Paulownia* tree cultivated at the Campus Klein-Altendorf, Germany. Copyright 2019, Georg Völkering.

*Paulownia* samples were cultivated in the Shanxi province in China. The authors used enzymatic hydrolysis for biomass pulping, resulting in a ratio of about 42% cellulose, 20% hemicellulose, and 20% lignin [110,111]. Prior to enzymatic hydrolysis, various pre-treatment methods had been investigated (i.e., using dilute acid, alkali, and alkali supported by ultrasonic pretreatment, with the last one being the most efficient method).

Ashori and colleagues studied Iranian-cultivated *Paulownia fortunei* L. fibers, with a specific focus on their chemical and morphological characteristics. Results showed that Iranian *Paulownia fortunei* L. consisted of holocellulose, alpha-cellulose (about 52%), lignin (about 25%), and further extractives (about 15%, isolated from basic media). In addition, the authors determined fiber characteristics (i.e., length, width, cell wall thickness). Of special interest and a focus of scientific investigations is the fibrous parenchyma, a promising raw material for paper of high density, due to the material having a high tensile strength [112].

Zahedi et al. studied the polypropylene (PP) filler additives used to reinforce the polymer bulk. The studied samples included canola, paulownia, and nanoclay fillers in varying concentrations (3 and 5 wt%). Compared to canola and nanoclay fillers, *Paulownia* particles significantly improved the mechanical properties of the studied composites. Transmission electron microscopy and X-ray diffraction were used to specify the final polymer morphology and filler dispersion within the polymer matrix [113].

Besides *Paulownia*, *Eucalyptus*, and *Pine* are further examples of fast-growing trees. Pertuzzatti et al. recently reported a study on thermomechanical densification influenced by process parameters of two different crops: *Eucalyptus grandis* and *Pinus elliottii* [114]. Samples of both woods showed comparable densities and mechanical strength. Most obviously, significant differences resulted from differences in crop composition. Thus, the *Eucalyptus* hemicellulose (in difference to *Pine*) mainly consists of xylose of a higher degree of acetylation, that is more susceptible to degradation. Nevertheless, *Eucalyptus* samples showed densities close to 1.0 $g \cdot cm^{-3}$ and improved mechanical properties (i.e., bending, hardness, impact resistance) after pre-treatment.

### 2.3.3. Cup Plants: *Silphium Perfoliatum*

Unlike *Miscanthus*, *Silphium perfoliatum* L. belongs to the class of perennial $C_3$ plants, with characteristic yellow flowers (Figure 6). Originally, it was cultivated in North America and then brought to Europe in the 18th century. Currently, *Silphium* crops are established and distributed all over the world, including North and South America (Chile, USA), Asia (China, Japan), and Europe (France, Switzerland, Romania, Czech Republic, Germany, Hungary, Poland, Austria, Russia), with the plants mainly being investigated as a raw material for biogas, biofuel, and chemical production. The

advantages of these plants as a raw material are the low maintenance requirements, optimal growth (even in arid conditions), and high yields (Figure 6) [115–117].

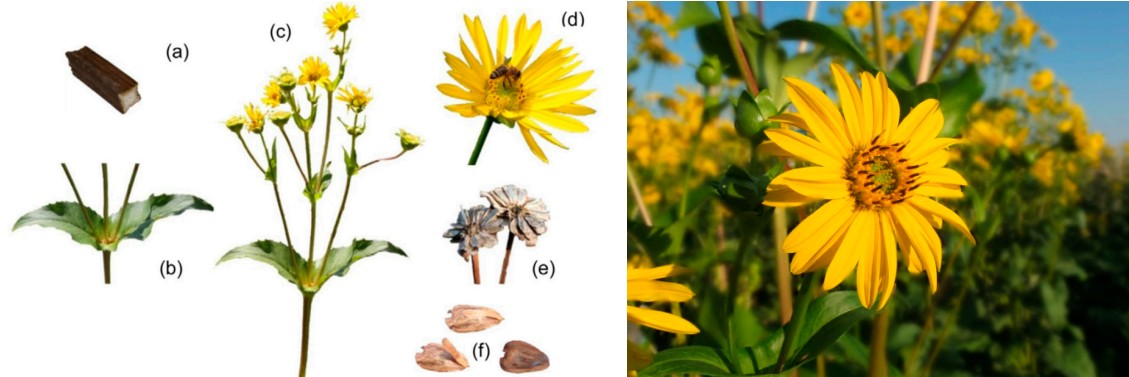

**Figure 6. Left**: *Silphium perfoliatum* L. (**a**) quadrangular stalk, (**b**) cup-shaped leaf axil, (**c**) branched stalk with flower buds and flower heads, (**d**) flower head with tubular and ligulate ray flowers, (**e**) mature inflorescence, and (**f**) fruits. Reproduced with permission from [111], Elsevier, 2019. **Right**: *Silphium perfoliatum* L. cultivated at the Campus Klein-Altendorf, Rheinbach, Germany. Copyright 2019, Georg Völkering.

*Silphium* crops are discussed as promising candidates for biogas production. According to Gansberger et al., the annual harvest yield can reach about 10 to 15 t per ha. Compared to maize, the biomethane production is 20% lower. However, so far there are a very limited number of studies and a lot of questions to be answered regarding the potential of these plants as lignocellulose feedstock. Thus, a seed technology must be developed, pathogen susceptibility has to be checked, and a suitable herbicide for weed management during the first cultivation year is most probably required [118].

The Lithuanian Research Centre for Agriculture and Forestry in Western Lithuania performed a field study reported by Šiaudinis and colleagues—the authors cultivated various perennial coarse-stemmed herbaceous energy plants, including mugwort (*Artemisia vulgaris* L.) and cup plant (*Silphium perfoliatum* L.). For their field trial, the authors used a two-factor design, including three levels of liming (not limed versus limed, using $CaCO_3$ in different concentrations) and nitrogen as the fertilizer in varying concentrations, to study the influence of these parameters on the cup plant dry matter productivity. Results showed that both fertilizer and lime significantly influence (decrease) the energy output and energy use efficiency [119]. So far, *Silphium perfoliatum* L. has been studied in detail regarding its utilization as an additive for food and pharmaceuticals and as raw materials for bioenergy and biofuel production [119,120].

In another study, Klímek and colleagues investigated the exploitation of agricultural crop residues as renewable sources for particleboard production. The following samples were studied: cup-plant (*Silphium perfoliatum* L.), sunflower (*Helianthusannuus* L.), and topinambour (*Helianthus tuberosus* L.). Particleboards of 600 kg/m$^3$ density were produced using different adhesives (methylene diphenyl diisocyanate, urea formaldehyde resin). Various physical and mechanical properties of the final boards were measured, including rupture modulus, thickness, swelling, and water absorption. Based on the obtained data, the authors concluded that agricultural crop residues can be used for particleboard and furniture production, meeting European standard EN 312 class P1 [121].

Papadopoulos et al. studied the exploitation of sunflower stalks as an alternative raw material for particleboards. As a pretreatment method, acetylation was conducted, to increase the thickness swelling (TS) of the boards. Thus, up to 19.7% weight gain could be obtained. Unfortunately, the introduction of acetyl functionalities resulted in a decrease in the internal bond strength. The authors concluded that a mixture of industrial wood chips and sunflower stalks might be appropriate to improve the particleboard specifications [122].

## 3. LCF Structure Analysis and Quality Control

### 3.1. Spectroscopic Data Processing Using Chemometric Methods for Biomass Analysis

Modern literature on the use of machine learning methods in chemical analysis (chemometrics) is, in general, quite extensive and diverse. In recent years, a large number of reviews have been published on individual methods and analyzed objects [123–127]. However, the number of studies using chemometric methods, against the background of the total number of analytical works, is still extremely small. Furthermore, even less work has been done that utilizes chemometrics for studying LCF. Iqbal and Lewandowski investigated the inter-annual variation in biomass yield and composition in a multi-genotype trial planted in southern Germany, focusing on climatic conditions (i.e., rainfall, temperature) and harvest dates [128]. Chemometric methods, such as multivariate regression analysis, were used to study correlations between harvesting time and rainfall. Boeriu et al. combined Fourier-transform infrared spectroscopy (FTIR) and principal component analysis (PCA) for the classification of the botanical origin of lignins [129]. Regression models (e.g., partial least squares, PLS) resulted in the accurate determination of phenolic hydroxyl groups, which could then be correlated to antioxidant capacity. Chen et al. used multivariate methods to process their experimental FTIR data obtained for various wood samples [130]. Results showed root-mean-square errors for all three LCF components, lignin, cellulose, and hemicellulose, of 1.51%, 0.96%, and 0.62%, respectively. Very recently, Lancefield et al. reported a study on lignin 3D structure analysis using attenuated total reflection (ATR)-FTIR analysis combined with PCA and PLS modeling. In addition, the obtained quantitative results were comparable to gel-permeation chromatography (GPC) and 2D heteronuclear single quantum coherence (HSQC) nuclear magnetic resonance (NMR) methods [131].

Thus, only classical chemometric methods have been used for the modeling of predominately FTIR data, leaving open many interesting topics for research. For example, nothing is known about the application of calibration transfer methods in LCF analysis, or the application of novel algorithms, such as independent component analysis (ICA), to improve existing chemometric models. The same is applicable for the complementary vibrational Raman spectroscopy, which gives important insights into a polymer's structure and its characteristics. These data also require multivariate methods for the data interpretation, due to overlapping peaks of polymers present in the data that cannot be interpreted without machine learning techniques.

Moreover, despite the obvious interest in multivariate modeling showed by some groups, there is no uniform methodology for applying machine learning methods in the analytical chemistry of LCF. It is also clear, however, that given the current level of automation, the amount of measured information, and throughput of analytical equipment, chemometrics should become an integral part of the analytical chemistry of natural polymers such as LCF.

The implementation of chemometrics can be helpful in different aspects of polymer analytical science. For example, up to now the determination of the molecular weight (MW), corresponding distribution (MWD), and polydispersity (PD) of natural macromolecular structures is usually performed via GPC (gel permeation chromatography) or SEC (size exclusion chromatography) using polystyrene (PS) or polymethyl methacrylate (PMMA) standards. Due to the complex and unique 3D structure of natural polymers (particularly lignin), the hydrodynamic volume usually differs significantly between standards and analytes [132]. Therefore, universal calibration or additional methods (i.e., osmometry, light scattering) have to be used in order to determine MW and polydispersity.

In general, experimental measurements can be replaced by multivariate models based on the modeling of spectroscopic data that possesses information about the molecular weight distribution of polymers (e.g., diffusion-ordered nuclear magnetic resonance, DOSY NMR). Other unexplored tasks include the evaluation of polymer linkages by using 2D NMR spectroscopy (HSQC, and heteronuclear multiple bond correlation, HMBC) and chemometrics, determination of the hydroxyl number, and total phenolic content, by spectroscopic techniques and others. Theoretical modeling can provide additional insights into the structure of lignin building blocks. Concerning existing instrumental techniques, no single analytical technique has been more comprehensively employed for the evaluation of LCF structure than NMR [21,23–25]. Yet, there is no example of multivariate techniques for resolving overlapping peaks in 1D and 2D NMR profiling of LCF, or multivariate modeling of specific $^{31}$P and $^{13}$C NMR profiles. Doing so will bring additional important insights into the polymer structure, and enable the construction of multivariate models for the determination of important LCF qualitative characteristics, such as crop genotype/phenotype and geographical origin.

X-ray fluorescence analysis (XRF) is a rarely used analytical tool for LCF, although it is an attractive method for performing inorganic elemental analysis [133]. Even if LCF is mainly composed from organic matter and light elements that cannot be detected directly with XRF, an application of chemometric techniques to the scattering XRF profile may provide valuable information on integral LCF parameters. In our ongoing research, XRF (in addition to spectroscopic methods) is used for quantitative biomass analysis, with respect to heavier elements that can be a marker of certain features and in combination with machine learning methods for ascertaining the type and origin of LCF. In Table 3, a variety of studies reporting the structure and composition analysis of LCF, using experimental analytical methods combined with multivariate data processing, are summarized [134–143].

Table 3. Chemometrics in LCF composition and structure analysis [134–144].

| Feedstock/Biomass Used | Data Analyzed | Experimental Methods Used | Chemometric Methods Used | References |
|---|---|---|---|---|
| **25 *Miscanthus* genotype samples**, i.e., *M. x giganteus, M. sinensis, M. Sacchari florussaccroflorus* | **Cell-wall composition** and lignin content of different *Miscanthus* stem and leaf samples. | FTIR and NMR spectroscopy | FTIR spectra processing using MatLab (regions 1900–800 cm$^{-1}$), transformation via Savitzky–Golay algorithm; PCA and Eigenvector PLS (version 7.0.3); statistics via Statistica (version 8.0 StatSoft, Tulsa, OK, USA). | [134] |
| *Miscanthus, Switchgrass, Reed Canary Grass* | **Element analysis** including N, S, P, Si, Cl, Na, K, Ca, and heating value | FTIR, NMR spectroscopy | Principal component analysis (PCA) | [135] |
| **Moso bamboo samples** from three sites in China: 15 culms of 5 physiological ages (1–5 years). Furthermore, samples obtained from 4 positions from each culm (base, middle, top, and middle node sections). 180 samples in total. | **Quantitative visualization** of lignocellulose components. | FTIR macro- and micro-spectroscopy | Partial least-squares regression (PLSR) and a Montecarlo sampling method (MSM) were used to establish the quantitative determination model of lignocelluloses. | [136] |
| **Dissolving pulp** | **Pulp composition** determined including pentosan, α-cellulose, viscosity, and brightness. | Wet chemical methods (TAPPI 2003-4), UV- and Fourier transformed near-infrared (FTNIR) spectroscopy | Pre-treatment by mean normalization, smoothing with moving average, Standard Normal Variate, Savitzky-Golay smoothing with first/second derivatives, and combinations. Raw and treated data processing using principal component regression (PCR) and partial least square regression (PLSR). | [137] |
| *Swietenia macrophylla King (Mahogany) and Eucalyptus hybrid (E. urophylla × E. camaldulensis).* | Determination of cellulose and lignin distribution in wood surfaces of *Swietenia macrophylla King (Mahogany)* and *Eucalyptus hybrid (E. urophylla × E. camaldulensis).* | Raman image spectroscopy (RIS) | The multivariate curve resolution-alternating least squares method is based on the bilinear model. The relative concentration maps were obtained by applying a multivariate curve resolution procedure. | [138] |
| **8 evaluated biomasses from greenhouse crop residues** (*Cucurbita pepo, Cucumis sativus, Solanum melongena, Solanum lycopersicum, Phaseoulus vulgaris, Capsicum annuum, Citrillus vulgaris Schrad, Cucumis melo*). | **Crop content prediction** of hemicellulose, cellulose (sugar content) and lignin. | 1D and 2D NMR spectroscopy (i.e., as 1H-1H TOCSY, 1H-13C HSQC, 1H-13C HMBC) | The experimental NMR data were processed using the PLS-DA model. The prediction of hemicellulose showed errors up to 22%, while for the other two components the errors are in all the cases below 1%. Discriminant buckets from a PLS-DA model combined with linear models provided a useful and rapid tool for the determination of cell wall composition. | [139] |
| **94 woodchip samples and 70 pellet samples** from different Italian power plants (March-May 2017 and February-May 2018). | **Prediction of different chemical-physical parameters** of woodchip and pellet samples, such as moisture content, net calorific value, ash content and gross calorific value of woodchip samples. | Vis-NIR spectroscopy with and without sample pre-treatment (i.e., grinding or stabilization at 40 °C for 24 h) | Visible NIR data were processed using partial least square regression to predict various chemical-physical parameters of wood-chips and pellets correlated to biofuel quality. Best results were obtained considering only the near IR region. | [140] |

**Table 3.** *Cont.*

| Feedstock/Biomass Used | Data Analyzed | Experimental Methods Used | Chemometric Methods Used | References |
|---|---|---|---|---|
| **Carob samples (flesh and seed) from seven different Mediterranean countries** (Cyprus, Greece, Italy, Spain, Turkey, Jordan and Palestine) | **Crop origin determination** via functional group analysis to be assigned to polysaccharides, lipids and proteins. | FTIR spectroscopy (recorded in transmittance mode) | Experimental data were processed statistically using multivariate chemometric techniques, including Principal Component Analysis (PCA), Cluster Analysis (CA), Partial Least Squares (PLS) and Orthogonal Partial Least Square-Discriminant Analysis (OPLSDA). Results confirmed that PCA was most useful to differentiate the studied carob samples, in particular the contribution of the geographical origin. | [141] |
| **Lignins from different origin**: i.e., soda-derived lignins (wheat straw and Sarkanda grass/wheat mixture), Organosolv lignin from maple/birch/poplar hard wood mixture, a pine-derived kraft lignin (Indulin AT) and an alkaline-isolated wheat straw-lignin. | **Crop origin determination** via functional group content analysis. | Fourier Transform Infrared (FTIR) and quantitative 31P NMR spectroscopy. | Principal component analysis and partial least squares regression analysis were used for data processing (Unscrambler® 7.6, Camo, Norway). PCA results showed differences of the studied lignin fractions. PLS could correlate 31P-NMR and FT-IR data with the chemical composition of lignin fractions. Authors reported a calibration model to predict the chemical parameters. PCA and the PLS model were validated using a new set of data (i.e., cross validation set). | [129,142,143] |
| **Lignins from different *Miscanthus* genotypes** including *M. giganteus, M. robustus and M. sinensis* harvested at different seasons and years, respectively, separated into leaf and stem | **Genotype composition**, monolignol ratio (G, H, versus S) and corresponding linkages. | FTIR, UV-Vis and NMR (HSQC) spectroscopy, GPC, Pyrolysis-GC/MS, composition analysis via NREL | Principal component analysis (PCA) | [26,27] |
| **54 different technical lignin samples** (including kraft, soda and organosolv pulping). | **Linkage abundance and molecular weight** characteristics of technical lignins | Attenuated Total Reflection-FTIR, gel-permeation chromatography (GPC) and nuclear magnetic resonance (NMR) for structure analysis of technical lignins. | Principal component analysis and partial least square modelling (using PLS_Toolbox v. 8.6, Eigenvector) in Matlab. Spectra were pre-processed using baseline correction, normalization and mean-centering. Results clearly showed similarities and deviations for the 54 lignins correlating to their botanic origin and pulping process (used for isolation). | [131] |

### 3.2. Chemometrics Used for Ligocellulose Feedstock Specification

Within the last five years, a tremendous number of LCF analysis studies have been reported, some of which include chemometric data processing (Table 3) [131,134]. For example, in 2014 Da Costa et al. reported an LCF cell-wall analysis study, including 25 *Miscanthus* genotypes of different developmental stages separated into stem and leaf portions. In detail, the authors combined mid-infrared spectroscopy with PCA in order to quantify the differences in cell-wall composition of stem and leaf-derived *Miscanthus* samples, which are in turn associated with different structural carbohydrates (Figure 7) [134].

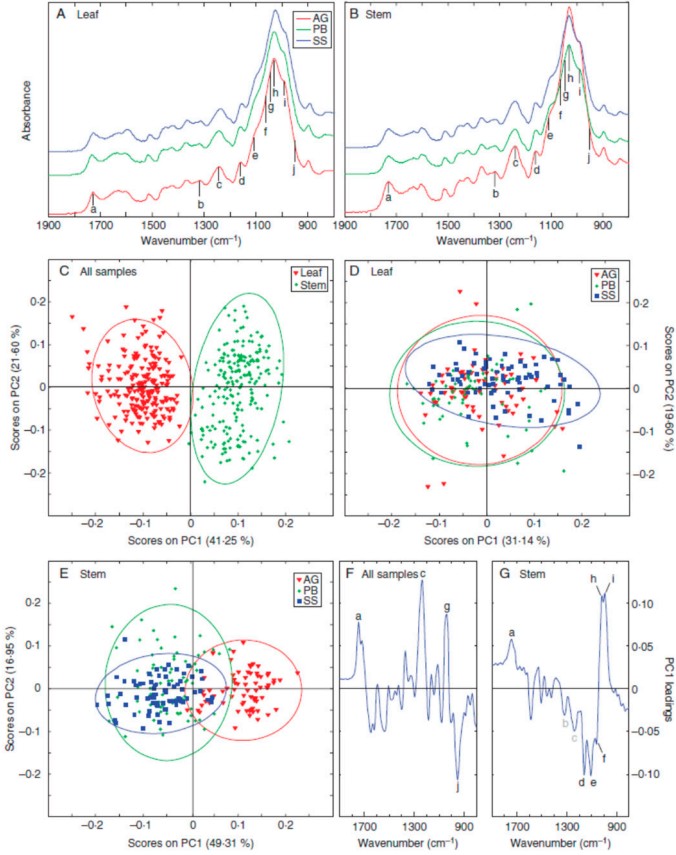

**Figure 7.** Mean Fourier-transform infrared spectroscopy (FTIR) spectra of (**A**) leaf and (**B**) stem samples of 25 *Miscanthus* genotypes, at three developmental stages in the range 1900–800 cm$^{-1}$. Plot of principal component one (PC1) and principal component two (PC2) scores for (**C**) all samples, (**D**) leaf samples, and (**E**) stem samples. PC1 loading plot for (**F**) all samples and (**G**) stem samples. Reproduced with permission from [134], Oxford University Press, 2019.

Schäfer et al. performed a study including a large number of *Miscanthus*, switchgrass, and reed canary grass samples, to investigate and compare the crop composition depending on origin and harvesting conditions [135]. In detail, ash, silicon, nitrogen, potassium, phosphorous, calcium, chloride, and sulfur content, and the heating value of the grasses were determined. Compared to switchgrass and reed canary grass, *Miscanthus* genotypes showed significantly lower ash contents (1.6% to 4.0%, compared to 1.9% to 10.5% and 11.5%, respectively).

Li and colleagues studied various moso bamboo samples, with regard to crop composition and ratio of cellulose versus hemicellulose and lignin, respectively. The samples (15 stalks of five ages) were collected from three different sites in China, including Jingning and Guangan counties, Sichuan Province. FTIR macro- and micro-spectroscopic imaging techniques, combined with chemometric

processing (using partial least-squares regression (PLSR) and Monte Carlo sampling to identify abnormal data), have been applied for quantitative analysis of moso bamboo crop composition [136].

Uddin et al. investigated the cellulose and hemicellulose content (in particular alpha-cellulose and pentosan), as well as properties such as pulp viscosity, of dissolving jute pulp, using wet chemical analysis and various spectroscopic methods (i.e., FTIR, UV-Vis) combined with chemometric data processing. The authors were able to develop a fast and reliable procedure to quantify the abovementioned biomass parameters of dissolving pulp, with the help of simple and fast spectroscopic nondestructive methods combined with chemometric data processing [137].

Colares and colleagues used Raman spectral imaging to specify the ratio of cellulose and lignin in surfaces of various trees (i.e., *Swietenia macrophylla* King, Mahogany/Eucalyptus hybrid, *E. urophylla* × *E. camaldulensis*). They used a multivariate 'curve resolution' procedure to calculate the relative concentration maps and simulate the Raman spectra for cellulose and lignins (finding good correlations with literature data). For all samples, the lignin concentration varied between 20% and 45% for the Eucalyptus samples and some higher values for the Mahogany tree (depending on the local origin). The authors aimed to show that Raman image spectroscopy combined with chemometric data analysis (i.e., multivariate curve resolution-alternating least squares MCR-ALS) is an appropriate tool for final specification of the cellulose/lignin ratio in Mahogany and Eucalyptus hybrids [138].

Aguilera-Saeza et al. very recently reported the structural analysis to determine the ratio of cellulose, hemicellulose, and lignin of eight different greenhouse crop residues, namely *Cucurbita pepo*, *Cucumis sativus*, *Solanum melongena*, *Solanum lycopersicum*, *Phaseolus vulgaris*, *Capsicum annuum*, *Citrullus vulgaris Schrad.,* and *Cucumis melo,* using chemometrics in NMR spectroscopy. In detail, the authors were able to specify correlations of metabolite profiles and cell wall composition using a PLS-DA (partial least square-discriminant analysis) and linear regression models (Figure 8) [139]. For reliability verification, composition analysis was also performed, according to the National Renewable Energy Laboratory (NREL) procedure, as control experiments.

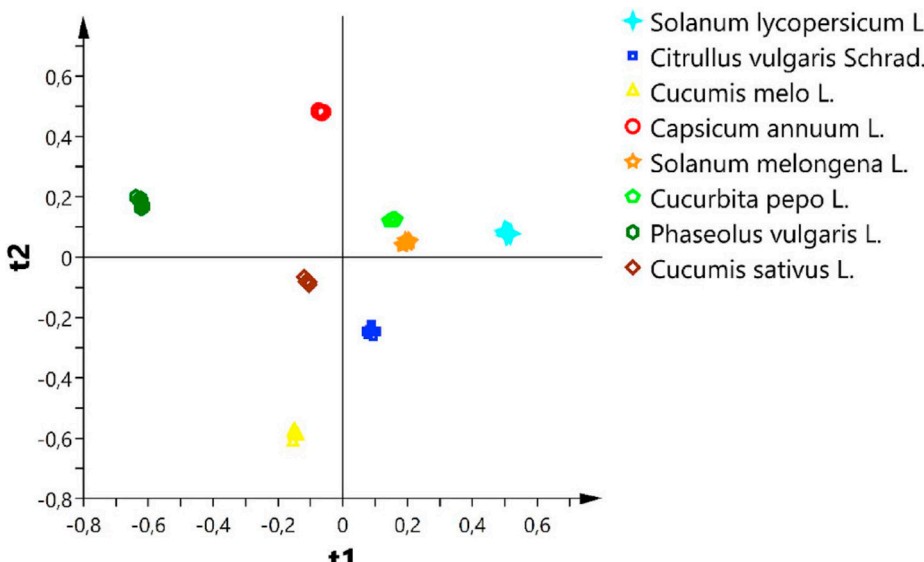

**Figure 8.** Partial least square-discriminant analysis (PLS-DA) scores plot derived from 80 $^1$H nuclear magnetic resonance (NMR) spectra for the eight different crop residue plant species evaluated. Reproduced with permission from [139], Elsevier 2019.

Woodchips and pellets of different plant species and origins have been studied by Mancini and colleagues in order to specify quantitative differences in their chemical composition. Methods used included wet chemical analysis and Vis-NIR (near infrared) spectroscopy, combined with chemometric data processing (i.e., PLS). The background for their study is the utilization of fast spectroscopy

methods for biofuel combustion quality, i.e., moisture content, net calorific value, and ash, according to EN ISO 17225. Chemometric data processing of the near infrared region delivered the best results [140].

Christou and colleagues investigated carob samples to specify their origin, using FTIR spectroscopy. With the help of PCA data processing, the authors were able to determine distinct groups, which could be assigned to the carob crop origin (Cyprus, Greece, Italy, Spain, Turkey, Jordan, and Palestine). In addition, chemometric methods, such as cluster analysis (CA), PLS, and Orthogonal Partial Least Square-Discriminant Analysis (OPLSDA), were applied, resulting in 95% confidence for origin specification (Figure 9) [141].

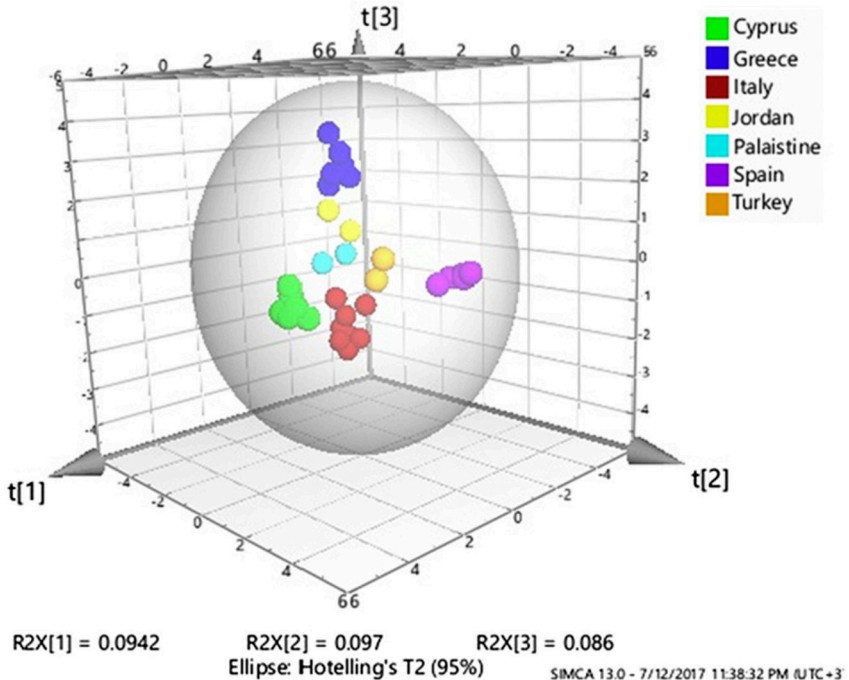

**Figure 9.** PLS plot from analysis on principal component analyses (PCAs) of 1st derivatives (2500–4000 cm$^{-1}$). Reproduced with permission from [141], Elsevier 2019.

Boeriua et al. studied the fractionation of different technical lignins using selective extraction in green solvents. Five samples were investigated, including two soda-derived lignins (wheat straw and a mixture consisting of Sarkanda grass/wheat from Greenvalue SA, Switzerland), one organosolv lignin (Alcell, obtained from maple/birch/poplar hard wood mixture, Repap Technologies Inc./USA), a pine-derived kraft lignin "Indulin AT" (MeadWestvaco/USA), and a wheat straw-based lignin from a mild alkaline process (Technical University Dresden/Germany). The chemical composition was determined via $^{31}$P NMR and corresponding data were processed using PCA, showing high heterogeneity (Figure 10) [142]. The different extraction procedures resulted in distinct deviations in the functional group content. Structural information regarding the *p*-hydroxyphenyl (H), guajacol (G) and syringol unit (G/H/S) ratio and aliphatic OH content was obtained from PLS models based on FTIR data.

Chen et al. investigated lignins of different origins using infrared spectroscopy to classify the botanical source. IR data were processed using PCA and partial least squares (PLS) regression to specify phenol-derived hydroxy groups, in order to draw correlation to the antioxidant activity of the lignins [130].

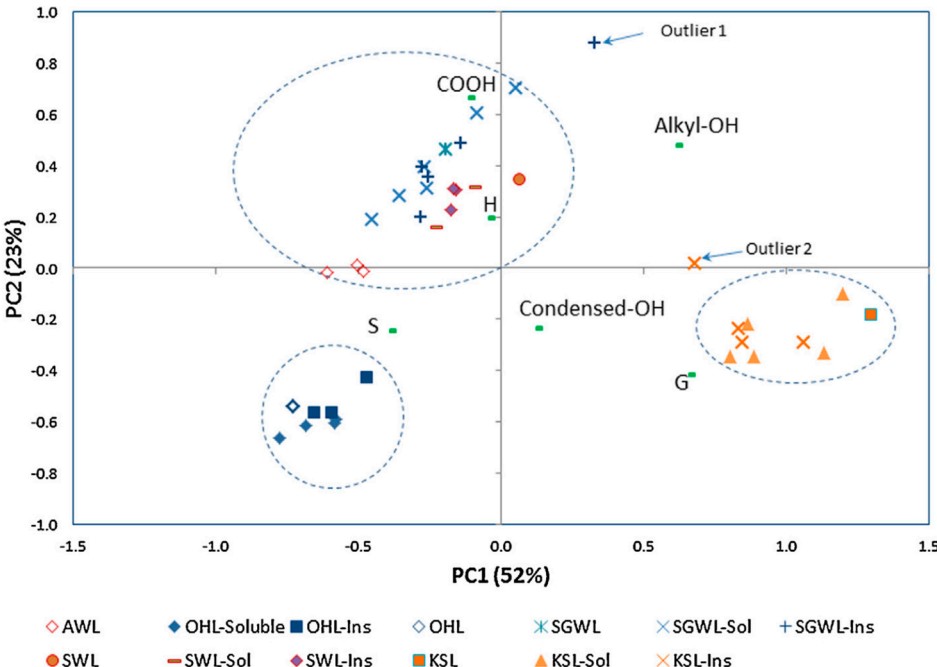

**Figure 10.** Plot of scores (samples) and loadings ($^{31}$P NMR chemical data) on principal component 1 (PC1) and 2 (PC2). Reproduced with permission from [142], Elsevier 2019.

Combining FTIR and multivariate data processing, Boeriu and Gosselink et al. examined a number of carob samples from seven different Mediterranean countries, using the first derivatives of the FTIR spectra, resulting in a confidence level of up to 95%. The contents of lignin, cellulose, and hemicellulose were determined. To do so, the authors processed a broad variety of input parameters, including wood species, resulting in root-mean-square errors of less than 1.51% [129,142,143].

Due to the fact that genotype and cultivation conditions significantly influence the 3D chemical structure of any crop, it is of importance to have access to specific plants. It should be emphasized that we do have unique access to well-defined LCF raw materials—crops cultivated at Campus Klein-Altendorf University Bonn, one of the largest field labs for *Miscanthus* cultivation in Europe (more than 30 genotypes), and further special biomasses, i.e., *Silphium perfoliatum, Paulownia*. In addition, there is also access to crops from specific harvesting seasons (i.e., September, December, April), specific years, and plant portions (leaf versus stem). Thus, in previous studies the correlation between crop genotype, harvesting time (year, season), plant portion and lignin amount, and 3D structure was investigated. For six different genotypes, the lignin content varies, as shown in Figure 11.

Based on this information, a decision can be made regarding the harvesting time (season) in order to obtain highest yields. In general, there are various advantages of *Miscanthus* cultivation: as a C$_4$ plant, the plants bind to four (instead of three) carbon atoms, resulting in an exceptional CO$_2$ fixation rate and high photosynthesis yields. Thus, *Miscanthus* crops are intensively studied for industrial exploitation, including lignin generation. Recently, we reported a systematic study showing strong correlations of the lignin structure with the *Miscanthus* genotype and plant portion (stem versus leaf) [26]. In detail, for lignins isolated via non-catalyzed organosolv, pulping the amount and linkages of the three monolignol building blocks (G, H, and S) was studied with different analytical methods (i.e., NREL protocol, FTIR, UV-Vis, HSQC-NMR, thermal gravimetric analysis (TGA), pyrolysis gas chromatography/mass spectrometry (GC/MS). The FTIR data have been processed using chemometric methods (i.e., principal component analysis). A comparison of beech wood and *Miscanthus* lignins could show that the *Miscanthus*-derived lignins showed lower molecular weight and narrow polydispersities (<1.5, compared to >2.5 for beech), most probably due to an increased homogeneity.

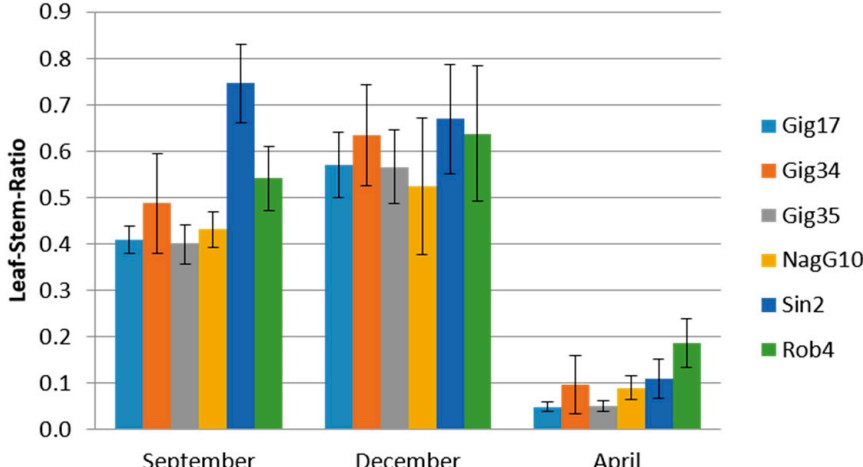

**Figure 11.** Leaf versus stem content of *Miscanthus* genotypes: *M. x giganteus* (Gig17, Gig34, Gig35), *M. nagara* (NagG10), *M. sinensis* (Sin2), and *M. robustus* (Rob4), harvested in September, December, and April, respectively (arranged to follow the seasonal order from autumn to spring). Reprinted from [27] under open access license.

The nature and ratio of different monolignol linkages has been studied in detail using heteronuclear single quantum correlation (HSQC) 2D-NMR. Results showed that leaves contain two-thirds of the G units, whereas in stems and mixtures the G content is rather low. Compared to G, H and S units were found to be highest in samples containing leaf and stem mixtures. Figure 12 shows the calculated ratio of the most abundant β-arylether (A) linkages (55–65%), followed by phenyl coumarane (B) and resinol (C) linkages. Stem-derived lignins mainly contain unsaturated esters (D) (ca. 30%). The concentration of residual carbohydrates was below the detection threshold, indicating the high purity of organosolv-derived lignins.

**Figure 12.** Lignin structure elements for heteronuclear single quantum correlation (HSQC) NMR signal assignment (A: β-O-4′ linkage, B: phenylcoumaran, C: resinol, D: β-unsaturated ester). Reprinted from [27] under open access license.

PCA processing of FTIR data from lignin samples was performed to determine the structural differences of lignins obtained from different *Miscanthus x giganteus* plant portions (stems, leaves, and their mixtures). Results are shown in Figure 13—the projections of IR spectra from lignin samples on the first three principal components (82% of variance). In particular, a differentiation of stem versus leaf-derived lignins was possible, since the aromatic in-plane deformation signals at 1160 cm$^{-1}$ do correspond to the monolignol substitution pattern (Figure 13) [27].

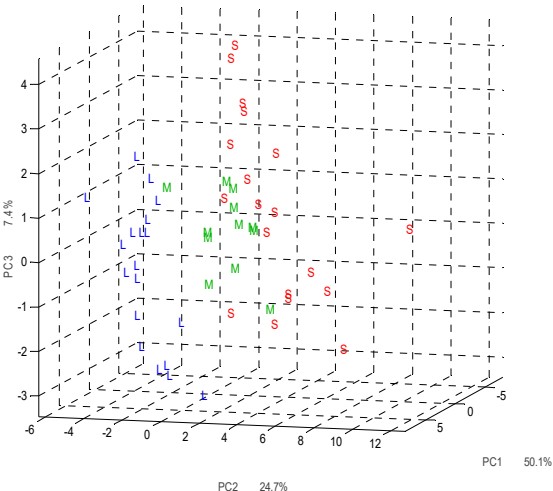

**Figure 13.** Multivariate data analysis of FTIR data using the principal component analysis (PCA). L: leaf-derived lignin, M: mixture-derived lignin, S: stem-derived lignin. Reprinted from [27] under open access license.

Lancefield et al. very recently reported a study including 54 lignin samples differing in origin and fractionation process. ATR-FTIR and NMR spectroscopy were used for structural analysis. The molecular weight and polydispersity were determined via gel permeation chromatography. All experimental data were processed using chemometric methods, i.e., PCA and PLS. Thus, molecular weight (number-average, Mn, and weight average, MW), as well as specific linkages (such as β-O-4, β-5, β-β′) were studied using PLS data processing of ATR-FTIR, GPC, and NMR, resulting in coefficients of determination (R2 Cal. > 0.85). Via PCA, soft and hard wood-derived lignins can be separated. Lancefield and colleagues then used the first derivative spectra, resulting in significantly improved resolution and sample separation (Figure 14) [131].

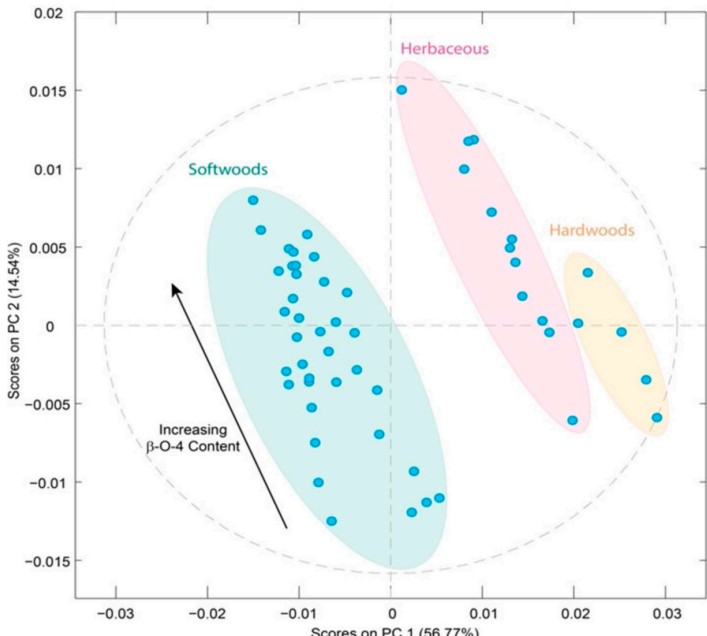

**Figure 14.** Principal component analysis plot of the 54 lignin samples used in this study. The FTIR spectra were pre-processed by applying baseline correction, 1st derivative transformation, normalization, and mean centering. Shading shows how the lignins are grouped according to their different botanical origins. The colored ellipses are intended for illustrative purposes only. Reproduced with permission from [131], Wiley-VCH 2019.

*3.3. Future Aspects Using Chemometrics for LCF Quality Control*

In many cases, a strong overlap of spectral bands, even in two-dimensional experimental data, hampers classical data interpretation. This situation leads to the application of alternative chemometric methods for signal modeling. Here, methods such as PLS, ridge regression, stepwise regression with variable selection, principle component regression, and independent component analysis are appropriate tools for chemometric modeling of experimental data for the determination of quantitative characteristics of natural polymers, as these methods have proven their versatility and effectiveness for complex samples [144,145]. Discriminant analysis algorithms, such as linear discriminant analysis (LDA), factorial discriminant analysis (FDA), and partial least squares-discriminant analysis (PLS-DA), are aimed at the construction of linear discriminant functions that maximize interclass dispersion and minimize intraclass variance by applying generalized decomposition. This arsenal of approaches is expected to be used for determining qualitative characteristics of biopolymers, such as their botanical origin. In addition, confusion matrices that compare information on the actual and predicted assignments of the samples for each particular group will be constructed to study the predictive ability of models. Common component and specific weights analysis (CCSWA) can be applied to simultaneously analyze different spectroscopic data sets (i.e., for processing of NMR, IR, Raman, XRF spectroscopy data) [146]. The possibility of transferring chemometric models between different spectrometers will be evaluated by calibration transfer methods, such as direct standardization (DS) and piece-wise direct standardization methods (PDS) [147]. Besides MATLAB packages, the possibility of Python3 can be explored for constructing multivariate models.

**Author Contributions:** A.A., M.B., X.T.D., S.E.K., J.R., equally contributed data to the manuscript. Y.M. and R.P. contributed resources, and M.L. did final English corrections. M.S. wrote and edited the paper.

**Funding:** This research was funded by BMBF program "IngenieurNachwuchs" projects "LignoBau" (03FH013IX4) and EFRE Infrastrukturförderung "Biobasierte Produkte" (EFRE0500035).

**Acknowledgments:** Bonn-Rhein-Sieg University/Graduate Institute for scholar ship (A.A., M.B.) and Erasmus-Mundus Avempace-II scholar ship (A.A.); Bonn-Rhein-Sieg University/TREE institute (S.E.K.); North Carolina State University in conjunction with the DAAD RISE Scholarship Program (M.L.). We thank Katharina Walbrück and Georg Völkering for providing the photographs of the plants (Figures 4–6).

**Conflicts of Interest:** The authors declare no conflict of interest.

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
