# Peer review of "Low-Input Crops as Lignocellulosic Feedstock for Second-Generation Biorefineries and the Potential of Chemometrics in Biomass Quality Control"

_applsci, doi:10.3390/app9112252_

Reviewer 1 Report

This paper, entitled Low-Input Crops as Lignocellulosic Feedstock for Second Generation Biorefineries and the Potential of Chemometrics in Biomass Quality Control, is a scholarly work and can increase knowledge and generate novelty in the field of lignocellulosic feedstock for second generation biorefineries. The paper is very interesting, the work described here is in the spotlight of current research works and the content is very relevant to Applied Sciences. The paper is well written and well related to exiting literature, as shown by the list of references.

The abstract and keywords are meaningful. The quality of the manuscript is high and the approach of the authors is relevant, showing the strength of the work (collection and compilation of data from several studies and papers).  I have few general comments about this paper:

- I think it would be nice if the authors could provide cost analysis (cost of production and transformation) for each class of lignocellulosic materials or feedstock, such as cost of production (€ or $ per ton or per hectar)

- The compilation of data in Table 2 is very nice and requires hard work, but from my point of view, the information are uncomplete. It would be interesting to provide, if available, the amount of feedstock used and the quantity of products generated for each products. I know that these data are not easy to collect.

- The authors should provide some information or consideration about environmental aspects of production for the lignocellulosic feedstock described in this paper, in terms of agronomy (chemical inputs such as fertilizers), economy (€ or $ per ha for the cultivation, harvesting, ...), logistics (cost of transportation), ...

- In the discussion section or conclusion, the authors should provide a discussion about the best indicators to use based on the structure analysis parameters or data used. Is there any possibility to define one or more best indicator allowing to facilitate choice of feedstock selection or lignocellulosic use? Is it possible to determine form this modelization and this work the best  lignocellulosic material or feedstock for one purpose or for one group of products? or at least to provide the characteristics required for one application or one product? Is it possible to provide a tool of decision?

I really appreciate the quality and the content of this paper. From my point of view, the paper could be published in Applied Sciences but requires some amendment as mentioned before. I recommend the following decision: ACCEPT AFTER MINOR REVISION.

Author Response

All changes are highlighted in the revised manuscript.

Reviewer-1

- I think it would be nice if the authors could provide cost analysis (cost of production and transformation) for each class of lignocellulosic materials or feedstock, such as cost of production (€ or $ per ton or per hectar)

Answer: Cost analysis data (including production and transportation costs) are difficult to obtain. In 2018, two lignin market studies have been published (see references 18,19). However, these studies are very expensive (and require copyright permission).   

·        Lignin Market—Forecasts from 2018 to 2023; ID: 4479455; Knowledge Sourcing Intelligence LLP: Noida, India, 2018; 104 p.

·        Lignin Market Analysis by Product (Lignosulphonates, Kraft Lignin, Organosolv Lignin) by Application (Macromolecules, Aromatics), by Region (North America, Europe, APAC, Central & South America, MEA), and Segment Forecasts, 2014–2025; ID: 4240413 Report; Grand View Research: San Francisco, CA, USA, 2017; 110p.

In general, it really was not the intension of our contribution to supply this kind of data - and we would like to apologize if this impression might have been created.

However, we added a new subsection (2.2) and cited seven additional references to summarize recently reported techno-economic analyses on LCF exploitation. In fact, there are a few data available, however, most of the studies focus biofuel and bioethanol production and corresponding costs (not lignin isolation which is the focus of our studies).

2.2   Reported Techno-Economic Analysis Studies

Currently, there are a number of techno-economic analysis studies reported including information about the economic value and environmental impact of single LCF products such as bioethanol. For example, in 2019 Da Silva et al. published an assessment of different LCF pretreatment processes for bioethanol production. Taking into account five different pretreatment procedures of lignocellulosic biomass, the authors found that diluted acid is the best choice for bioethanol production with an economic value of 39.2 M$ per year and an environmental impact of 83.9 kt CO2 per year [62]. Patel et al. tried to quantify the production cost of biodiesel from agricultural waste, a comparative assessment recently reported [63]. Also in 2019, Albashabsheh et al. published their study on „mobile pelleting“, a procedure applied to improve and optimize lignocellulosic biomass-to-biofuel supply chains. In particular, the authors investigated mobile pelleting machines (MPM) to minimize logistic costs and to find out at which point mobile densification becomes economically attractive. Therefore, they included about 20 different input parameters like type and price of biomass, densification and transport costs, storage capacity and number of MPMs available [64]. A similar approach was reported by Srivastava et al. in 2019 to analyze costs for biofuel production [65].

In her PhD thesis, Karkee investigated the optimization and cost analysis of LCF supply chains. Considering corn stover as a by-product of grain production, the gate price of the biomass feedstock varies from $75 Mg-1 to $97 Mg-1 (depending on different factors such as farm size, transport distance and stover yield) [66]. Furthermore, the costs for harvesting and transport have been determined for different feedstocks (i.e. switchgrass). Quantification models were used considering the number of machines, farm size and biomass yields. Zhao et al. reported a Chinese market techno-economic analysis for the production of bioethanol. In particular, the pretreatment using dilute acids and an enzymatic hydrolysis were studied for corn stover biomass. Using two different models, the authors calculated the plant-gate price for bioethanol and reported it to be $4.68–$6.05/gal following a biochemical conversion pathway. Thus, at this price point, ethanol from lignocellulose biomass is still unable to compete with ethanol from fossil resources [67].

In their techno-economic analysis study reported in 2011, Gnansounou et al. comprehensively reviewed data for ethanol production from lignocellulosic feedstocks. They could identify and quantify some key parameters influencing the production costs like type and composition of feedstock and its farm-gate price, conversion efficiency, the ethanol plant size and the extent of investment costs using three different types of cost management system, whereby the most significant contribution to the overall lignocellulosic bioethanol production costs is the biomass cost [68].

·         Da Silva, A.R.G.; Giuliano, A.; Errico, M.; Rong, B.G.; Barletta, D. Economic value and environmental impact analysis of lignocellulosic ethanol production: assessment of different pretreatment processes. Clean Technol. Env. Policy 2019, 21, 637–654, doi:10.1007/s10098-018-01663-z.

·         Patel, M.; Oyedun, A.O.; Kumar, A.; Gupta, R. What is the production cost of renewable diesel from woody biomass and agricultural residue based on experimentation? A comparative assessment. Fuel Proc. Technol. 2019, 191, 79–92, doi:10.1016/j.fuproc.2019.03.026.

·         Albashabsheh, N.T.; Heier Stamm, J.L. Optimization of lignocellulosic biomass-to-biofuel supply chains with mobile pelleting. Transport. Res. Part E 2019, 122, 545–562, doi:10.1016/j.tre.2018.12.015.

·         Srivastava, N.; Kharwar, R.K.; Mishra, P.K. Cost Economy Analysis of Biomass-Based Biofuel Production. In New and Future Developments in Microbial Biotechnology and Bioengineering. From Cellulose to Cellulase: Strategies to Improve Biofuel Production, Elsevier, Amsterdam, The Netherlands, 2019, Pages 1-10, doi:10.1016/B978-0-444-64223-3.00001-1.

·         Karkee, A. Optimization and cost analysis of lignocellulosic biomass feedstocks supply chains for biorefineries. PhD thesis, 2016, Iowa State University, Capstones, U.S., doi:10.31274/etd-180810-4601.

·         Zhao, L.; Zhang, X.; Xu, J.; Ou, X.; Chang, S.; Wu, M. Techno-Economic Analysis of Bioethanol Production from Lignocellulosic Biomass in China: Dilute-Acid Pretreatment and Enzymatic Hydrolysis of Corn Stover. Energies 2015, 8, 4096-4117, doi:10.3390/en8054096.

·         Gnansounou, E.; Dauriat, A. Technoeconomic Analysis of Lignocellulosic Ethanol, in: Biofuels. Alternative Feedstocks and Conversion Processes. In Biofuels: Alternative Feedstocks and Conversion Processes, Elsevier, Amsterdam, The Netherlands, 2011, pp. 123-148, doi:10.1016/B978-0-12-385099-7.00006-1.

- The compilation of data in Table 2 is very nice and requires hard work, but from my point of view, the information are uncomplete. It would be interesting to provide, if available, the amount of feedstock used and the quantity of products generated for each products. I know that these data are not easy to collect.

Answer: Both, feedstock and “products” differ (very much) for each of the company listed in Table 2. To our best knowledge, detailed data on feedstock amount and “product quantity” might be specified in annual reports. To collect these data, extensive literature search is required (similar to data included in the lignin market studies, see first question).

- The authors should provide some information or consideration about environmental aspects of production for the lignocellulosic feedstock described in this paper, in terms of agronomy (chemical inputs such as fertilizers), economy (€ or $ per ha for the cultivation, harvesting, ...), logistics (cost of transportation), ...

Answer: For the plants discussed in our manuscript (Miscanthus, Paulownia, Silphium), we tried to give a few information on cultivation conditions (including fertilizer utilization, harvesting costs) citing the following references, e.g.:

For Miscanthus cultivation:

·         Clifton‐Brown, J.; Harfouche, A.; Casler, M.D. Breeding progress and preparedness for mass‐scale deployment of perennial lignocellulosic biomass crops switchgrass, miscanthus, willow and poplar. GCB Bioenergy. 2019, 11, 118–151, doi: 10.1111/gcbb.12566.

·         Iqbal, Y.; Lewandowski, I. Lignocellulosic Energy Grasses for Combustion, Production, and Provision. In: Energy from Organic Materials (Biomass), 1st ed.; Springer: New York, NY, 2019; pp. 89-99. Online ISBN 978-1-4939-7813-7.

For Paulownia cultivation:

·         Pereira, J.S.; Landsberg, J.J. (Eds.) Biomass Production by Fast-Growing Trees. Print ISBN 978-94-010-7557-2, Springer, Dordrecht. doi:10.1007/978-94-009-2348-5.

·         Ye, X.; Chen and Y. Kinetics Study of Enzymatic Hydrolysis of Paulownia by Dilute Acid, Alkali, and Ultrasonic-assisted Alkali Pretreatments. Biotechnol. Bioproc. Eng. 2015, 20, 242-248, doi:10.1007/s12257-014-0490-x.

·         Ayrilmis, N. and Kaymakci, A. Fast growing biomass as reinforcing filler in thermoplastic composites: Paulownia elongate wood. Ind. Crop. Prod. 2013, 43: 457-464, , doi: 10.1016/j.indcrop.2012.07.050.

·         Cheng, J., Sun, Y.; Chen, Y. Optimization of dilute acid pretreatment of Paulownia for the production of bioethanol by respond surface methodology. Adv. Mat. Res. 2012, 250-553.

·         Ashori, A. and Nourbakhsh, A. Studies on Iranian cultivated paulownia–a potential source of fibrous raw material for paper industry. Eur. J. Wood Prod. 2009, 67, 323–327, doi:10.1007/s00107-009-0326-0.

For Silphie cultivation:

·         Conrad, M.; Biertümpfel, A.; Vetter, A. Durchwachsene Silphie (Silphiumperfoliatum L.) – von der Futterpflanze zum Koferment F.N.R. Gülzower Fachgespräche. In: Presented at the 2nd Symposium Energiepflanzen, FachagenturNachwachsende Rohstoffe, Gülzow, Germany, pp. 281–289, 2009.

·         Aurbacher, J.; Benke, M.; Formowitz, B.; Glauert, T.; Heiermann, M.; Herrmann, C.; Idler, C.; Kornatz, P.; Nehring, A.; Rieckmann, C.; Rieckmann, G.; Reus, D.; Vetter,A.; Vollrath, B.; Wilken, F.; Willms, M. Energiepflanzen für Biogasanlagen (Broschüre No. 553). Fachagentur Nachwachsende Rohstoffe e.V. (Ed.), Rostock, Germany, 2012, pp. 1-84.

·         Stolzenburg, K.; Monkos, A. Erste Versuchsergebnisse mit der Durchwachsenen Silphie (Silphium perfoliatum L.) in Baden-Württemberg. Landwirtschaftliches Technologiezentrum Augustenberg (Ed.), Karlsruhe, Germany, 2012.

·         Gansberger, M.; Montgomery, L.F.R.; Liebhard, P. Botanical characteristics, crop management and potential of Silphium perfoliatum L. as a renewable resource for biogas production: A review. Ind. Crops Prod. 2015, 63, 362–372, doi: 10.1016/j.indcrop.2014.09.047.

- In the discussion section or conclusion, the authors should provide a discussion about the best indicators to use based on the structure analysis parameters or data used. Is there any possibility to define one or more best indicator allowing to facilitate choice of feedstock selection or lignocellulosic use? Is it possible to determine form this modelization and this work the best lignocellulosic material or feedstock for one purpose or for one group of products? or at least to provide the characteristics required for one application or one product? Is it possible to provide a tool of decision?

Answer: So far, “best” indicators for LCF analysis are not specified. All references cited mainly focus and use similar analysis methods (such as FTIR, NMR, SEC etc.), however, the goal of the studies differs: some focus biofuel and/or bioethanol production, others lignin isolation and so on. Similar to cost analysis data, “best” indictors are difficult to be specified and always depend on the final application.

For lignin isolation and structural analysis, FTIR combined with multivariate data analysis seems to be an appropriate approach to gain information correlated to botanic origin. So, we tried to discuss these aspects partially in section “3.2. Chemometrics Used for Ligocellulose Feedstock Specification” citing most recent literature data (see also Table 3).

Reviewer 2 Report

The review manuscript was well written. The authors presented systematic studies of selected lignocellulose feedstocks such as Miscanthus, Paulownia, Eucalyptus, and Silphium perfoliatum. Particularly, a list of different pilot plants and industrial production sites was provided. Different chemometric methods to analyze and quantify the biomass components and lignin structural characteristics were reviewed and discussed carefully within the text. A table of various techniques were provided as well. This review paper will be a very useful source and reference for researchers in the field. I recommend publishing this review paper after correcting and changing some minor points:

1.      In the Featured Application(s) (lines 22-25), there is a typo in the numbering.

2.      In Figure 1 (lines 57-58), the authors should remove the “Figure 5” in the description of Ref [5].

3.      The authors should check and correct the discussion in lines 277-279 and lines 416-422. The authors repeated the same statement. Also please specify which reference you mentioned in lines 277-279? Is it from Chen et al. Ref [123] or Christou et al. Ref [136]? (see lines 416-422).  

Author Response

All changes are highlighted in the revised manuscript.

Reviewer-2:

1.      In Figure 1 (lines 57-58), the authors should remove the “Figure 5” in the description of Ref [5].

Answer: This figure is reprinted (with copyright permission). We asked the publisher (Springer Nature) to get the original figure (without legend).

3.      The authors should check and correct the discussion in lines 277-279 and lines 416-422. The authors repeated the same statement. Also please specify which reference you mentioned in lines 277-279? Is it from Chen et al. Ref [123] or Christou et al. Ref [136]? (see lines 416-422).   

Answer: In lines 277-279 we wrote: “However, the number of studies using chemometric methods, against the background of the total number of analytical works, is still extremely small. Furthermore, even less work has been done that utilizes chemometrics for studying LCF.”  Stating that there are only a very few studied reported so far using chemometric methods for LCF structure analysis.

Both references (Chen et al. and Christou et al.) are cited and discussed separately: see Chen et al. [130] and Christou et al. [144]. All reference numbers changed due to additional citations (see above, reviewer-1).

Furthermore, a typing error in author name (Boeriou, reference [142] was corrected and [143] has been added. Boeriu, C.G.; Bravo, D.; Gosselink, R.J.A.; van Dam, J.E.G. Characterisation of structure-dependent functional properties of lignin with infrared spectroscopy. Ind. Crops Prod. 2004, 20, 205–218, doi:10.1016/j.indcrop.2004.04.022.